# Efficacy and Safety of a Parenteral Nutrition Program for Patients with *RAS* Wild-Type Metastatic Colorectal Cancer Administered First-Line Cetuximab Plus Chemotherapy: A Propensity Score Matching Study

**DOI:** 10.3390/nu15132971

**Published:** 2023-06-30

**Authors:** Yu-Tang Chang, Chou-Chen Chen, Shih-Ching Chang, Yu-Yao Chang, Bo-Wen Lin, Hong-Hwa Chen, Yao-Yu Hsieh, Hung-Chih Hsu, Meng-Che Hsieh, Feng-Che Kuan, Chih-Chien Wu, Wei-Chen Lu, Yu-Li Su, Yi-Hsin Liang, Joe-Bin Chen, Shuan-Yuan Huang, Ching-Wen Huang, Jaw-Yuan Wang

**Affiliations:** 1Division of Pediatric Surgery, Department of Surgery, Kaohsiung Medical University Hospital, Kaohsiung Medical University, Kaohsiung 80708, Taiwan; 890300@ms.kmuh.org.tw; 2Department of Surgery, Faculty of Medicine, College of Medicine, Kaohsiung Medical University, Kaohsiung 80708, Taiwan; 3Department of Surgery, Taichung Veterans General Hospital, Taichung 40705, Taiwan; ccchen24615237@gmail.com; 4Division of Colon and Rectal Surgery, Department of Surgery, Veterans General Hospital, Taipei 11217, Taiwan; scchang3@ym.edu.tw; 5Department of Colorectal Surgery, Changhua Christian Hospital, Changhua 50006, Taiwan; 177176@cch.org.tw (Y.-Y.C.); 140111@cch.org.tw (S.-Y.H.); 6Department of Post-Baccalaureate Medicine, College of Medicine, National Chung Hsing University, Taichung 40227, Taiwan; 7Department of Surgery, National Cheng Kung University Hospital, Tainan 70457, Taiwan; wen276@gmail.com; 8Division of Colon and Rectal Surgery, Department of Surgery, Chang Gung Memorial Hospital, Kaohsiung 83301, Taiwan; hs168@adm.cgmh.org.tw; 9Division of Hematology and Oncology, Shuang Ho Hospital, Taipei Medical University, New Taipei City 23561, Taiwan; alecto39@gmail.com; 10Division of Hematology and Oncology, Department of Internal Medicine, School of Medicine, College of Medicine, Taipei Medical University, Taipei 11031, Taiwan; 11Division of Hematology-Oncology, Department of Internal Medicine, Chang Gung Memorial Hospital at Linkou, Taoyuan 33305, Taiwan; b12117@adm.cgmh.org.tw; 12College of Medicine, Chang Gung University, Taoyuan 33305, Taiwan; 13Division of Hematology-Oncology, Department of Internal Medicine, E-Da Hospital, I-Shou University, Kaohsiung 84001, Taiwan; 14Department of Hematology and Oncology, Chang Gung Memorial Hospital, Chiayi 61363, Taiwan; it905180@gmail.com; 15Division of Colorectal Surgery, Department of Surgery, Kaohsiung Veterans General Hospital, Kaohsiung 81362, Taiwan; pauleoswu@gmail.com; 16School of Medicine, National Yang Ming Chiao Tung University, Taipei 30010, Taiwan; 17Department of Oncology, National Taiwan University Hospital Yunlin Branch, Yunlin 64041, Taiwan; coste@ms16.hinet.net; 18Division of Hematology and Oncology, Department of Internal Medicine, Chang Gung Memorial Hospital, Kaohsiung 33305, Taiwan; yolisu@cgmh.org.tw; 19Department of Oncology, National Taiwan University Hospital, Taipei 10002, Taiwan; yihsinliang@ntu.edu.tw; 20Department of Surgery, Chung Shan Medical University Hospital, Taichung 40201, Taiwan; crsmyway@gmail.com; 21Division of Colorectal Surgery, Department of Surgery, Kaohsiung Medical University Hospital, Kaohsiung 80756, Taiwan; 22Graduate Institute of Clinical Medicine, College of Medicine, Kaohsiung Medical University, Kaohsiung 80708, Taiwan; 23Center for Cancer Research, Kaohsiung Medical University, Kaohsiung 80708, Taiwan; 24Pingtung Hospital, Ministry of Health and Welfare, Pingtung 90054, Taiwan

**Keywords:** cetuximab, *RAS* wild-type, metastatic colorectal cancer, supplementary home parenteral nutrition

## Abstract

Malnutrition is a common problem in patients with metastatic colorectal cancer (mCRC) receiving targeted therapy plus chemotherapy, resulting in severe toxicity and decreased survival rates. This retrospective study employing propensity score matching (PSM) examined the efficacy and safety of a supplemental home parenteral nutrition (HPN) program for patients with *RAS* wild-type mCRC receiving cetuximab plus chemotherapy. This retrospective nationwide registry study included data from 14 medical centers/hospitals across Taiwan, and the data period ranged from November 2016 to December 2020. Patients with *RAS* wild-type mCRC receiving cetuximab plus chemotherapy as their first-line therapy were included and divided into HPN and non-HPN program groups. HPN was initiated based on patient-specific factors, such as baseline nutritional status, treatment-related toxicities, and comorbidities. Clinical outcomes were evaluated using response to therapy, duration of response (DoR), progression-free survival (PFS), and overall survival (OS). This study recruited 758 patients, of whom 110 and 648 were included in the HPN and non-HPN program groups, respectively. After 1:3 PSM, the data of 109 and 327 patients from the HPN and non-HPN program groups were analyzed, respectively. The HPN program group had a higher metastasectomy rate (33.9% vs. 20.2%, *p* = 0.005), and longer duration of treatment and DoR than the non-HPN program group (13.6 vs. 10.3 and 13.6 vs. 9.9 months, *p* = 0.001 and < 0.001, respectively). The HPN program group tended to have a longer median PFS (18.2 vs. 13.9 months, *p* = 0.102). Moreover, we noted a significant improvement in the median OS in the same group (53.4 vs. 34.6 months, *p* = 0.002). Supplemental HPN programs may be recommended for select patients with mCRC receiving targeted therapy plus chemotherapy to improve oncological outcomes.

## 1. Introduction

Colorectal cancer (CRC) is a prevalent cancer, with an estimated 1.9 million cases and 915,880 deaths reported globally in 2020 [1]. In Taiwan, CRC has become the most commonly diagnosed cancer and the third leading cause of cancer-related deaths since 2006, with over 18,400 new cases and an incidence rate of 51.14 per 100,000 individuals in 2020 [2]. Furthermore, a considerable proportion (18.9%) of patients with CRC in Taiwan have metastatic CRC (mCRC) [2], highlighting the need for effective and advanced therapies to prolong such patients’ survival and improve their quality of life [3].

According to the American Cancer Society, the 5-year survival rate of patients with mCRC is approximately 13% [4]. The survival rate of patients with mCRC depends on several factors, including the extent of cancer, the patient’s age and overall health, and the treatment they receive [4]. Treatment options for CRC include surgery, chemotherapy, targeted therapy, and immunotherapy. The choice of treatment depends on the location and extent of cancer and various individual factors. *RAS* wild-type mCRC refers to a type of mCRC that does not have mutations in the *RAS* gene [5]. Cetuximab plus chemotherapy substantially improved the overall response rate (ORR), progression-free survival (PFS), and overall survival (OS) in patients with *RAS* wild-type mCRC [3,6,7]. However, this combination therapy can cause considerable gastrointestinal toxicity, including diarrhea, nausea, and vomiting, exacerbating malnutrition and weight loss [8].

Malnutrition and weight loss are common problems in patients with mCRC and considerably affect their recovery process and ability to tolerate the side effects of cancer therapy [8]. In patients with a severe case of malnutrition, a feeding tube may be necessary to provide nutrition directly to the body. According to the European Society for Clinical Nutrition and Metabolism, parenteral nutrition (PN) may be recommended when patients with cancer have severe malnutrition or are unable to absorb nutrients through the digestive system due to factors such as surgery [9]. Costanzo et al. reported that PN significantly prolonged survival by 3 months in patients with metastatic cancer [10]. When patients with advanced cancer are unable to receive adequate nutrients enterally or orally while continuing their anticancer therapy, home parenteral nutrition (HPN) may be an important component of their care plan [11,12,13].

The routine use of PN support in patients with incurable cancer is generally not recommended [9]. The evidence supporting the benefits of PN intervention in cancer patients is limited. However, nutrition support continues to be prescribed for patients with cancer. Malignant disease represents a significant proportion of cases requiring HPN, accounting for approximately half of all cases in a large series [14]. Despite the limited evidence, individualized assessments should be conducted for each patient to determine the most suitable approach to nutrition support based on their specific circumstances and needs.

In the context of mCRC patients with *RAS* wild-type receiving cetuximab plus chemotherapy, the impact of supplemental HPN remains uncertain. Therefore, we conducted a retrospective study using propensity score matching (PSM) to investigate the efficacy and safety of treatment with and without a supplemental HPN program in this patient population. The objective of this study was to assess whether the use of supplemental HPN could improve treatment outcomes for mCRC patients undergoing cetuximab plus chemotherapy.

## 2. Patients and Methods

### 2.1. Study Design and Patient Population

A retrospective nationwide registry study was conducted between November 2016 and December 2020 across 14 medical centers/hospitals in Taiwan, and it focused on patients with *RAS* wild-type mCRC who received cetuximab plus chemotherapy as first-line therapy. Treatment regimens were previously reported [15]. Patients received cetuximab plus FOLFIRI (folinic acid, fluorouracil, and irinotecan) or FOLFOX (folinic acid, fluorouracil, and oxaliplatin) on day 1 of each 14-day treatment cycle. Treatment was continued until disease progression was noted, unacceptable toxic effects occurred, a complete response was achieved, surgical resection became possible, or the patient or physician decided to discontinue treatment. Cetuximab therapy was halted if any patient developed a grade 3 or 4 allergic or hypersensitivity reaction.

Enrolled patients were categorized into HPN and non-HPN program groups. In the supplemental HPN program group, the use of HPN was individualized on the basis of patient-specific factors, such as baseline nutritional status, treatment-related toxicities, and comorbidities. The supplemental HPN program was initiated to improve nutritional status if a patient’s nutritional risk index was ≤97.5 during treatment [16]. In addition, PN was used to manage treatment-related toxicities, such as diarrhea and mucositis, which can affect oral intake and nutritional status. Close monitoring and assessment of nutritional status were essential to ensure the timely and appropriate adjustment of PN therapy. It is important to note that patients with an estimated life expectancy of less than three to six months were not offered with the supplemental HPN program. This decision was made considering their limited life expectancy and the potential burden of initiating long-term nutritional support. The present study was approved by local ethics committees (KMUHIRB-E(I)-20200036) and conducted in accordance with guidelines contained in the Declaration of Helsinki of 1975, as revised in 1996.

### 2.2. Clinical Outcome Assessment

Responses to therapy were assessed using computed tomography based on the Response Evaluation Criteria in Solid Tumors (version 1.1) [17]. Tumor markers, such as carcinoembryonic antigen (CEA) and carbohydrate antigen 19-9 (CA19-9), were measured to evaluate treatment response and disease behavior. The primary endpoint was metastasectomy, objective response rate (ORR), duration of treatment (DoT), duration of response (DoR), disease control rate (DCR), and adverse events. PFS and OS were secondary endpoints. PFS was calculated from the treatment start date until tumor progression or death. OS was calculated from the treatment start date to the time of death due to any cause. For patients who were alive at the final analysis, data on survival were censored at the last contact date.

### 2.3. Statistical Analysis

To minimize potential selection bias, we used PSM to match compatible groups [18]. The PSM caliper matching method (matching ratio, 1:3) was used to match patients in the HPN program group with those in the non-HPN program group, with consideration for patients’ demographic characteristics (age and sex), clinical cancer stage (including T and N stages), tumor location (distance from the anal verge), Eastern Cooperative Oncology Group performance status, and body mass index. All data were fully anonymized before analysis. The baseline characteristics of included patients and response data are presented as *n* (%) or median and interquartile range, as appropriate. Clinical and demographic data were compared using appropriate statistical tests, including Pearson’s chi-squared test, Fisher’s exact test, or Student’s *t*-test. PFS and OS are expressed as median values with 95% confidence intervals (CIs) and estimated using the Kaplan–Meier method. The log-rank test was used to compare time-to-event distributions. Statistical significance was set at *p* < 0.05. All statistical analyses were performed using SPSS version 21.0 (IBM Corp, Armonk, NY, USA).

## 3. Results

### 3.1. Study Populations and Baseline Characteristics

This study included 758 patients with mCRC who received cetuximab plus chemotherapy as the first-line treatment. Among them, 110 and 648 patients were included in the HPN and non-HPN program groups, respectively. In the HPN program group, HPN was administered in 13 (11.8%) patients. The median duration of HPN administration was 94 (range, 6–298) days. After PSM was applied, 109 patients from the HPN program group and 327 from the non-HPN program group were included in the analysis. Table 1 summarizes the baseline characteristics of the patients before (*n* = 758) and after (*n* = 436) PSM.

Before PSM was applied, significant differences in age (*p* = 0.036) and the number of metastatic sites (*p* = 0.008) were observed between the groups. However, following PSM, no differences were noted between the groups after adjustment for confounding variables (all *p* > 0.05).

### 3.2. Efficacy

Table 2 summarizes the oncological outcomes of the patients before and after PSM. The HPN program group had a higher response rate to therapy than the non-HPN program group (*p* = 0.013). Moreover, the HPN program group had a higher metastasectomy rate than the non-HPN program group (33.9% vs. 20.6%, *p* = 0.005). However, no significant differences in the ORR or DCR were noted between the groups (*p* = 0.928 and 0.754, respectively). Moreover, the HPN program group had a longer DoT and DoR than the non-HPN program group (13.6 vs. 10.3 and 13.6 vs. 9.9 months, *p* = 0.001 and <0.001, respectively).

### 3.3. Adverse Events

Table 3 and Table 4 summarize the adverse events before and after PSM, respectively. After PSM was applied, the HPN program group was revealed to have a higher risk of hematologic and non-hematologic adverse events (*p* < 0.001 and *p* < 0.001, respectively). The HPN program group had higher rates of high-grade hematologic adverse events, such as anemia (27.5% vs. 3.1%), neutropenia (8.3% vs. 0.6%), and febrile neutropenia (22.0% vs. 2.1%), than the non-HPN program group (*p* < 0.001; Table 4).

### 3.4. Survival Rate

Figure 1 displays the Kaplan–Meier survival curves for the HPN and non-HPN program groups before and after PSM. After PSM was applied, the median PFS for the HPN program group was 18.2 (95% CI: 14.7–21.7) months compared with 13.9 (95% CI: 12.5–15.3) months for the non-HPN program group. PFS at 5 years was 14.4% in the HPN program group and 15.8% in the non-HPN program group; the 5-year PFS did not significantly differ between the groups (*p* = 0.102).

After PSM was applied, we noted a 5-year OS rate of 45.2% and median OS of 53.4 (95% CI: 36.3–70.4) months in the HPN program group. In the non-HPN program group, the OS was 22.8% and the median OS was 34.6 (95% CI: 31.3–38.0) months. The HPN program group had a significantly longer OS than the non-HPN program group (53.4 vs. 34.6 months, *p* = 0.002).

## 4. Discussion

Cetuximab is a monoclonal antibody that targets the epidermal growth factor receptor (EGFR), which is often overexpressed in different types of cancer, including CRC [19]. *RAS* genes are frequently mutated in CRC and play a crucial role in cell signaling pathways [20]. Cetuximab is only effective in patients with *RAS* wild-type tumors. In patients with *RAS* wild-type mCRC, cetuximab has been observed to be effective when used in combination with chemotherapy as the first-line treatment [3,6]. Various clinical trials have examined the efficacy and safety of cetuximab plus chemotherapy in the first-line treatment of *RAS* wild-type mCRC [15,21]. These studies have consistently indicated that compared with chemotherapy alone, cetuximab plus chemotherapy more effectively enhanced OS, PFS, and ORR.

One of the most significant trials was the CRYSTAL trial, which enrolled 1198 patients with untreated mCRC [21]. Patients were randomly assigned to receive either FOLFIRI chemotherapy or FOLFIRI plus cetuximab. The findings indicated that the addition of cetuximab to FOLFIRI improved OS by 1.5 months, PFS by 1.4 months, and ORR by 10%. Another large trial was the FIRE-3 trial [15], which enrolled 592 patients with untreated mCRC. Patients were randomly assigned to receive FOLFIRI plus cetuximab or FOLFIRI plus bevacizumab. The results revealed that the addition of cetuximab to FOLFIRI improved OS by 3.7 months, PFS by 1.6 months, and ORR by 14%. A recent study indicated that the median PFS in patients with mCRC receiving first-line bevacizumab and FOLFIRI was extended from 10 months to 14 months through the identification of the *UGT1A1* polymorphism and the administration of a subsequent dose escalation of irinotecan [22]. In the present study, after PSM, we observed that the use of the HPN program in conjunction with cetuximab plus chemotherapy as the first-line treatment led to an 18.8-month improvement in OS in patients with mCRC.

Various factors can affect the efficacy and safety of first-line cetuximab plus chemotherapy in patients with *RAS* wild-type mCRC. For example, the location of the primary tumor (i.e., in the colon or rectum) can affect the response to cetuximab-based therapy. Patients with left-sided colon tumors typically had better outcomes than those with right-sided tumors. A large study conducted in Taiwan, which retrospectively analyzed 1583 patients with mCRC, determined that the median PFS and OS were longer in the EGFR left-sided colon group (9.8 and 34.3 months, respectively) than in the EGFR right-sided colon group (5.8 and 13.8 months, respectively) [23]. In addition, the performance status of patients, which reflects their ability to perform daily activities, may affect their treatment response. Patients with satisfactory performance status tended to have better outcomes than those with poor performance status [24]. Furthermore, patients’ characteristics, such as age, sex, and comorbidities, can affect the safety of cetuximab plus chemotherapy. Older patients may be more susceptible to adverse events, and those with preexisting medical conditions may have a higher risk of complications [25]. To mitigate potential selection bias, PSM was used in this study to match compatible groups.

Cetuximab plus chemotherapy can cause adverse events in the gastrointestinal system, such as diarrhea, nausea, and vomiting, leading to decreased food intake and nutrient absorption [8]. This can further exacerbate poor appetite, malnutrition, and weight loss in patients. Other potential complications of cetuximab plus chemotherapy include skin rash, infusion reactions, and electrolyte imbalance [26]. Moreover, chemotherapeutic drugs used to treat mCRC can cause myelosuppression, leading to low blood cell counts [27]. The toxicity profile can also affect treatment efficacy because patients who experience severe toxicity may require a dose reduction or treatment interruption, which can compromise treatment effectiveness. In the present study, a longer treatment duration was associated with a higher percentage of adverse events. However, supplemental HPN may improve the ability of patients to tolerate the side effects of systemic treatment, enabling them to receive the treatment for an extended duration and potentially leading to better oncological outcomes.

Malnutrition is a common problem in patients with mCRC and can adversely affect recovery time and the ability to tolerate the side effects of cancer therapy [8]. The use of the first-line cetuximab plus chemotherapy for *RAS* wild-type mCRC can worsen malnutrition due to the adverse effects of the treatment. Moreover, chemotherapy may damage the digestive tract lining, hindering nutrient absorption and contributing to malnutrition. Patients with malnutrition may have a weaker immune system, decreased tolerance to chemotherapy, and an increased risk of treatment-related complications. Supplemental HPN can provide adequate nutritional support during aggressive cancer treatment, potentially leading to better tolerance of chemotherapy and cetuximab. However, the use of PN in this setting should be individualized and based on patient-specific factors, such as baseline nutritional status, treatment-related toxicities, and comorbidities [28]. Careful monitoring of adverse events and prompt management of complications can ensure treatment safety and improve patient outcomes.

Limited clinical data are available that specifically address the timing of PN in the context of cetuximab plus chemotherapy for mCRC. According to clinical practice, as outlined in the present study, PN might be initiated early during chemotherapy or targeted therapy, ideally 7 to 10 days before the initiation of therapy. The early introduction of PN can prevent treatment-related toxicities, such as diarrhea and mucositis, which affect oral intake and nutritional status. Furthermore, the early initiation of PN can help maintain adequate nutritional status and support immune function, thus possibly improving treatment outcomes and survival.

Metastasectomy, the surgical removal of metastatic tumors in patients with mCRC, is a crucial treatment modality for select patients with limited metastatic disease because it can improve outcomes and prolong survival [29,30]. In the present study, the use of supplemental HPN was associated with a higher rate of metastasectomy. This finding may be attributed to improved nutritional status and better tolerance to therapy, which may have enabled patients to undergo the surgical resection of metastatic tumors. Furthermore, the improved clinical outcomes in the HPN program group may have allowed them to continue treatment for a longer period, leading to a longer overall treatment duration. These patients may have more effectively tolerated the treatment and thus received more cycles of chemotherapy and cetuximab, resulting in a better treatment response, higher survival rates, and longer survival times. Recently, Yeh et al. reported that patients receiving ≥7 cycles of chemotherapy plus targeted therapy exhibited a higher response rate, lower progression rate, and higher survival rate (all *p* < 0.05); however, adverse events did not significantly differ among groups [31]. In the current study, a median OS of 53.4 months was achieved using cetuximab plus chemotherapy in conjunction with the HPN program in the neoadjuvant setting in patients with mCRC.

HPN can improve the quality of life, nutritional status, and functional outcomes in patients with advanced cancer; however, it does not treat the underlying cause of malnutrition [32]. Furthermore, although HPN enables patients to live a more normal life outside the hospital, it is a high-risk and complex therapy associated with both immediate and long-term complications, particularly in patients with advanced cancer [11,12,13]. In terms of safety, HPN can be associated with several risks, such as infection, electrolyte imbalance, glucose intolerance, liver dysfunction, and catheter-related complications [32]. Given the limited life expectancy and the patient’s overall condition, the risks associated with nutritional support may outweigh any potential benefits. The greatest benefit was observed in patients who received HPN for 3 months, although patients receiving HPN for 1 or 2 months also demonstrated significant improvement [33]. Therefore, the decision to use HPN should be made by a multidisciplinary team who consider factors such as the patient’s overall health status, treatment goals, and the potential risks and benefits of the relevant intervention [34].

While nutritional support can offer benefits to specific patients with mCRC, its use should be approached with caution and based on individual patient evaluations [35]. Indiscriminate administration of PN to well-nourished or mildly malnourished patients is not advised due to insufficient evidence demonstrating its overall effectiveness in these patient groups. However, for severely malnourished patients or those at high risk of malnutrition due to factors such as impaired intake, gastrointestinal obstruction, or severe treatment side effects, PN may be necessary to maintain or restore their nutritional status. In such cases, the potential benefits of nutritional support outweigh the associated risks.

It is important to consider the appropriateness of nutritional intervention for patients with mCRC based on their disease prognosis and potential for a curative outcome or a long disease-free period after cancer treatment. Patients who have a potentially curable disease or can anticipate a significant period free from cancer after treatment may be best suited for consideration for nutritional intervention. Conversely, in situations where mCRC patients are terminally ill and have a limited life expectancy ranging from weeks to a few months, the routine use of PN support is not recommended. During this stage, the focus of care shifts towards palliative care, prioritizing comfort and symptom management rather than pursuing aggressive nutritional interventions.

This study has several limitations that should be considered. First, the retrospective nature of this study may limit the accuracy and reliability of the collected data. Second, the study was non-randomized and non-controlled. Thus, the absence of a comparison group limits the ability to directly attribute outcomes to the intervention (cetuximab plus chemotherapy with or without HPN). Third, the small sample size of the study may reduce the statistical power and generalizability of the findings. Fourth, the study only included patients with *RAS* wild-type mCRC who received cetuximab plus chemotherapy in the first-line setting; the study population may not represent the entire population of patients with mCRC. Finally, the use of HPN in the supplemental HPN program group was individualized on the basis of patient-specific factors; this may have introduced confounding factors and affected outcomes.

## 5. Conclusions and Perspective

The implementation of a supplemental HPN program may improve the efficacy and safety of first-line cetuximab plus chemotherapy in patients with *RAS* wild-type mCRC. HPN can improve nutritional status, enabling patients to receive treatment for a longer duration and reducing treatment interruptions. In addition, HPN can prevent adverse events and enhance treatment effectiveness, leading to improved patient survival. Thus, the supplemental HPN program can be recommended for select patients with mCRC receiving targeted therapy plus chemotherapy to potentially improve oncological outcomes.

## Figures and Tables

**Figure 1 nutrients-15-02971-f001:**
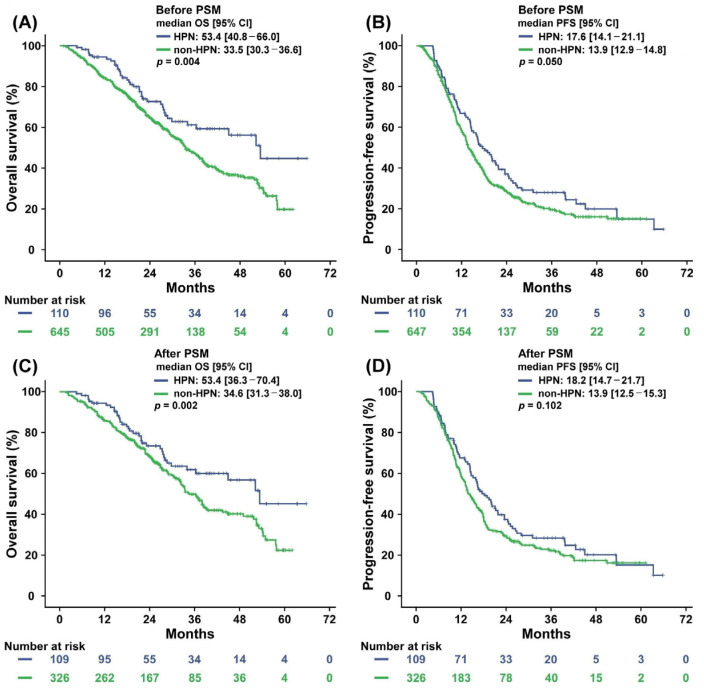
Survival outcomes of patients with *RAS* wild-type mCRC before and after propensity score matching. Kaplan–Meier survival plots of (**A**) OS and (**B**) PFS before propensity score matching and (**C**) OS and (**D**) PFS after propensity score matching. HPN, HPN program group; Non-HPN, non-HPN program group.

**Table 1 nutrients-15-02971-t001:** Baseline characteristics of the *RAS* wild-type patients with mCRC between supplemental HPN program group and non-supplemental HPN program group before (*n* = 758) and after (*n* = 436) propensity score matching.

Baseline Data		Overall				*p*	After Match (1:3)				*p*
		HPN (*N* = 110)		Non-HPN (*N* = 648)			HPN (*N* = 109)		Non-HPN (*N* = 327)		
Gender						0.725					>0.999
	Male	73	(66.4%)	441	(68.1%)		72	(66.1%)	216	(65.4%)	
	Female	37	(33.6%)	207	(31.9%)		37	(33.9%)	111	(33.6%)	
Age (years)						0.100					0.315
	Median (IQR)	64.4	(55.4, 73.1)	61.6	(51.5, 69.8)		63.9	(55.4, 73.1)	64.7	(52.1, 71.7)	
Age (years)						0.036					>0.999
	<65	55	(50.0%)	393	(60.6%)		55	(50.5%)	165	(50.9%)	
	≥65	55	(50.0%)	255	(39.4%)		54	(49.5%)	162	(49.1%)	
ECOG performance status						0.209					>0.999
	0 + 1	108	(98.2%)	616	(95.1%)		107	(98.2%)	321	(98.2%)	
	2	2	(1.8%)	31	(4.8%)		2	(1.8%)	6	(1.8%)	
	Unknown	0	(0.0%)	1	(0.1%)		0	(0.0%)	0	(0.0%)	
Primary lesion site						0.856					0.910
	Left-sided	100	(90.9%)	581	(89.7%)		99	(90.8%)	294	(89.9%)	
	Right-sided	10	(9.1%)	62	(9.6%)		10	(9.2%)	31	(9.5%)	
	Unknown/Both	0	(0.0%)	5	(0.7%)		0	(0.0%)	2	(0.6%)	
Synchronous/metachronous						0.122					0.774
	Synchronous	61	(55.5%)	407	(62.8%)		61	(56.0%)	187	(56.7%)	
	Metachronous	49	(44.5%)	237	(36.6%)		48	(44.0%)	138	(41.8%)	
	Unknown	0	(0.0%)	4	(0.6%)		0	(0.0%)	2	(1.5%)	
*BRAF* genotyping						0.331					0.276
	Wild type	108	(98.2%)	439	(67.8%)		107	(98.2%)	207	(62.7%)	
	Mutant type	1	(0.9%)	15	(2.3%)		1	(0.9%)	7	(2.1%)	
	Unknown	1	(0.9%)	194	(29.9%)		1	(0.9%)	113	(34.2%)	
Metastatic sites						0.173					0.292
	Liver	39	(35.5%)	249	(38.4%)		39	(35.8%)	145	(44.3%)	
	Lungs	16	(14.5%)	52	(8.0%)		16	(14.7%)	32	(9.8%)	
	Liver + lungs	8	(7.3%)	56	(8.7%)		8	(7.3%)	18	(5.5%)	
	Others	47	(42.7%)	291	(44.9%)		46	(42.2%)	132	(40.4%)	
Number of metastatic sites						0.008					>0.999
	1	73	(66.4%)	337	(52.0%)		73	(67.0%)	219	(67.0%)	
	≥2	36	(32.7%)	295	(45.5%)		36	(33.0%)	108	(33.0%)	
	Unknown	1	(0.9%)	16	(2.5%)		0	(0.0%)	0	(0.0%)	
Serum CEA level before treatment						0.455					0.617
	<5 ng/mL	33	(30.0%)	147	(22.7%)		33	(30.3%)	78	(23.9%)	
	≥5 ng/mL	77	(70.0%)	407	(62.8%)		76	(69.7%)	206	(63.0%)	
	Unknown	0	(0.0%)	94	(14.5%)		0	(0.0%)	43	(13.1%)	

mCRC, metastatic colorectal cancer; HPN, supplemental HPN program group; Non-HPN, non-supplemental HPN program group; IQR, interquartile ranges; ECOG, Eastern Cooperative Oncology Group; CEA, carcinoembryonic antigen.

**Table 2 nutrients-15-02971-t002:** The comparison of efficacy between supplemental HPN program group and non-supplemental HPN program group before (*n* = 758) and after (*n* = 436) propensity score matching.

			Overall				*p*	After Match (1:3)				*p*
			HPN (*N* = 110)		Non-HPN (*N* = 648)			HPN (*N* = 109)		Non-HPN (*N* = 327)		
Response to cetuximab in first-line treatment							0.063					0.013
	CR		3	(2.7%)	68	(10.5%)		3	(2.8%)	45	(13.8%)	
	PR		60	(54.6%)	307	(47.4%)		60	(55.0%)	148	(45.3%)	
	SD		30	(27.3%)	163	(25.2%)		29	(26.6%)	90	(27.5%)	
	PD		12	(10.9%)	78	(12.0%)		12	(11.0%)	33	(10.1%)	
	Not evaluable/Unknown		5	(4.6%)	32	(4.9%)		5	(4.6%)	11	(3.4%)	
Metastasectomy rate			37	(33.6%)	123	(19.0%)	0.001	37	(33.9%)	66	(20.2%)	0.005
Metastatic site resection							0.001					0.005
	No resection		73	(66.4%)	516	(79.6%)		72	(66.1%)	255	(78.0%)	
	Resection		37	(33.6%)	123	(19.0%)		37	(33.9%)	66	(20.2%)	
		R0 resection	26	(23.6%)	100	(15.4%)		26	(23.9%)	55	(16.8%)	
		R1 resection	10	(9.1%)	20	(3.1%)		10	(9.2%)	10	(3.1%)	
		R2 resection	1	(0.9%)	3	(0.5%)		1	(0.9%)	1	(0.3%)	
	Unknown		0	(0.0%)	9	(1.4%)		0	(0.0%)	6	(1.8%)	
ORR							0.865					0.928
	CR + PR		63	(57.3%)	375	(57.9%)		63	(57.8%)	193	(59.0%)	
	SD + PD		42	(38.2%)	241	(37.2%)		41	(37.6%)	123	(37.6%)	
DCR							0.724					0.754
	CR + PR + SD		93	(84.5%)	538	(83.0%)		92	(84.4%)	283	(86.5%)	
	PD		12	(10.9%)	78	(12.0%)		12	(11.0%)	33	(10.1%)	
Survival							0.011					0.013
	Yes		72	(65.5%)	339	(52.3%)		72	(66.1%)	171	(52.3%)	
	No		38	(34.5%)	309	(47.7%)		37	(33.9%)	156	(47.7%)	
DoT (month)			13.5	(7.0, 23.3)	9.8	(6.0, 14.4)	<0.001	13.6	(7.0, 23.7)	10.3	(6.4, 14.7)	0.001
DoR (month)			15.6	(9.8, 30.0)	7.8	(4.2, 15.4)	<0.001	15.6	(9.8, 30.0)	9.9	(4.8, 16.9)	<0.001

HPN, supplemental HPN program group; Non-HPN, non-supplemental HPN program group; CR, complete response; PR, partial response; SD, stable disease; PD, progressive disease; CI, confidence interval; ORR, objective response rate; DCR, disease control rate; DoT, duration of treatment; DoR, duration of response.

**Table 3 nutrients-15-02971-t003:** Adverse events of 758 *RAS* wild-type mCRC patients receiving cetuximab in first-line treatment between supplemental HPN program group and non-supplemental HPN program group before propensity score matching.

		All Grades				*p*	Grade 1–2				*p*	Grade ≥3				*p*
		HPN (*N* = 110)		Non-HPN (*N* = 648)			HPN (*N* = 110)		Non-HPN (*N* = 648)			HPN (*N* = 110)		Non-HPN (*N* = 648)		
Hematologic (overall)		101	(91.8%)	314	(48.5%)	<0.001	96	(87.3%)	298	(46.0%)	<0.001	30	(27.3%)	27	(4.2%)	<0.001
	Anemia	86	(78.2%)	214	(33.0%)	<0.001	77	(70.0%)	210	(32.4%)	<0.001	9	(8.2%)	4	(0.6%)	<0.001
	Neutropenia	89	(80.9%)	156	(24.1%)	<0.001	65	(59.1%)	134	(20.7%)	<0.001	24	(21.8%)	22	(3.4%)	<0.001
	Febrile neutropenia	4	(3.6%)	8	(1.2%)	0.083	1	(0.9%)	6	(0.9%)	>0.999	3	(2.7%)	2	(0.3%)	0.024
	Thrombocytopenia	11	(10.0%)	37	(5.7%)	0.088	11	(10.0%)	34	(5.2%)	0.052	0	(0.0%)	3	(0.5%)	>0.999
Non-hematologic (overall)		110	(100%)	507	(78.2%)	<0.001	110	(100%)	502	(77.5%)	<0.001	9	(8.2%)	49	(7.6%)	0.824
	Skin reaction	104	(94.5%)	374	(57.7%)	<0.001	102	(92.7%)	353	(54.5%)	<0.001	2	(1.8%)	21	(3.2%)	0.559
	Paronychia	17	(15.5%)	147	(22.7%)	0.087	17	(15.5%)	147	(22.7%)	0.087	0	(0.0%)	0	(0.0%)	-
	Abdominal pain	2	(1.8%)	57	(8.8%)	0.011	2	(1.8%)	54	(8.3%)	0.010	0	(0.0%)	3	(0.5%)	>0.999
	Diarrhea	42	(38.2%)	148	(22.8%)	0.001	40	(36.4%)	140	(21.6%)	0.001	2	(1.8%)	8	(1.2%)	0.645
	Nausea	70	(63.6%)	216	(33.3%)	<0.001	66	(60.0%)	215	(33.2%)	<0.001	4	(3.6%)	1	(0.2%)	0.002
	Vomiting	69	(62.7%)	146	(22.5%)	<0.001	65	(59.1%)	141	(21.8%)	<0.001	4	(3.6%)	5	(0.8%)	0.030
	Fatigue	87	(79.1%)	251	(38.7%)	<0.001	87	(79.1%)	239	(36.9%)	<0.001	0	(0.0%)	12	(1.9%)	0.232
	Infusion reaction	2	(1.8%)	11	(1.7%)	>0.999	2	(1.8%)	11	(1.7%)	>0.999	0	(0.0%)	0	(0.0%)	-
	Infection	6	(5.5%)	16	(2.5%)	0.086	5	(4.5%)	13	(2.0%)	0.107	1	(0.9%)	3	(0.5%)	0.468
	ALT increased	18	(16.4%)	60	(9.3%)	0.024	17	(15.5%)	59	(9.1%)	0.041	1	(0.9%)	1	(0.2%)	0.270
	AST increased	16	(14.5%)	66	(10.2%)	0.175	15	(13.6%)	65	(10.0%)	0.258	1	(0.9%)	1	(0.2%)	0.270
	Bilirubin increased	2	(1.8%)	24	(3.7%)	0.313	2	(1.8%)	22	(3.4%)	0.559	0	(0.0%)	2	(0.3%)	>0.999
	Creatinine increased	16	(14.5%)	28	(4.3%)	<0.001	16	(14.5%)	24	(3.7%)	<0.001	0	(0.0%)	4	(0.6%)	>0.999
	Hypomagnesemia	1	(0.9%)	16	(2.5%)	0.492	1	(0.9%)	16	(2.5%)	0.492	0	(0.0%)	0	(0.0%)	-

mCRC, metastatic colorectal cancer; HPN, supplemental HPN program group; Non-HPN, non-supplemental HPN program group; ALT, alanine aminotransferase; AST, aspartate transaminase.

**Table 4 nutrients-15-02971-t004:** Adverse events of 436 *RAS* wild-type mCRC patients receiving cetuximab in first-line treatment between supplemental HPN program group and non-supplemental HPN program group after propensity score matching.

After Propensity Score Matching (1:3)		All Grades				*p*	Grade 1–2				*p*	Grade ≥3				*p*
		HPN (*N* = 109)		Non-HPN (*N* = 327)			HPN (*N* = 109)		Non-HPN (*N* = 327)			HPN (*N* = 109)		Non-HPN (*N* = 327)		
Hematologic (overall)		100	(91.7%)	150	(45.9%)	<0.001	95	(87.2%)	144	(44.0%)	<0.001	30	(27.5%)	10	(3.1%)	<0.001
	Anemia	85	(78.0%)	100	(30.6%)	<0.001	76	(69.7%)	98	(30.0%)	<0.001	9	(8.3%)	2	(0.6%)	<0.001
	Neutropenia	88	(80.7%)	81	(24.8%)	<0.001	64	(58.7%)	74	(22.6%)	<0.001	24	(22.0%)	7	(2.1%)	<0.001
	Febrile neutropenia	4	(3.7%)	2	(0.6%)	0.037	1	(0.9%)	2	(0.6%)	>0.999	3	(2.8%)	0	(0.0%)	0.016
	Thrombocytopenia	11	(10.1%)	20	(6.1%)	0.196	11	(10.1%)	18	(5.5%)	0.098	0	(0.0%)	2	(0.6%)	>0.999
Non-hematologic (overall)		109	(100.0%)	273	(83.5%)	<0.001	109	(100%)	272	(83.2%)	<0.001	9	(8.3%)	36	(11.0%)	0.457
	Skin reaction	103	(94.5%)	229	(70.0%)	<0.001	101	(92.7%)	214	(65.4%)	<0.001	2	(1.8%)	15	(4.6%)	0.261
	Paronychia	17	(15.6%)	98	(30.0%)	0.003	17	(15.6%)	98	(30.0%)	0.003	0	(0.0%)	0	(0.0%)	-
	Abdominal pain	2	(1.8%)	39	(11.9%)	0.001	2	(1.8%)	36	(11.0%)	0.002	0	(0.0%)	3	(0.9%)	0.576
	Diarrhea	42	(38.5%)	96	(29.4%)	0.078	40	(36.7%)	91	(27.8%)	0.084	2	(1.8%)	5	(1.5%)	>0.999
	Nausea	70	(64.2%)	130	(39.8%)	<0.001	66	(60.6%)	129	(39.4%)	<0.001	4	(3.7%)	1	(0.3%)	0.015
	Vomiting	69	(63.3%)	100	(30.6%)	<0.001	65	(59.6%)	96	(29.4%)	<0.001	4	(3.7%)	4	(1.2%)	0.113
	Fatigue	87	(79.8%)	168	(51.4%)	<0.001	87	(79.8%)	158	(48.3%)	<0.001	0	(0.0%)	10	(3.1%)	0.073
	Infusion reaction	2	(1.8%)	6	(1.8%)	>0.999	2	(1.8%)	6	(1.8%)	>0.999	0	(0.0%)	0	(0.0%)	-
	Infection	6	(5.5%)	10	(3.1%)	0.248	5	(4.6%)	7	(2.1%)	0.180	1	(0.9%)	3	(0.9%)	>0.999
	ALT increased	18	(16.5%)	27	(8.3%)	0.015	17	(15.6%)	26	(8.0%)	0.021	1	(0.9%)	1	(0.3%)	0.439
	AST increased	16	(14.7%)	37	(11.3%)	0.398	15	(13.8%)	36	(11.0%)	0.445	1	(0.9%)	1	(0.3%)	0.439
	Bilirubin increased	2	(1.8%)	9	(2.8%)	0.594	2	(1.8%)	8	(2.4%)	>0.999	0	(0.0%)	1	(0.3%)	>0.999
	Creatinine increased	16	(14.7%)	15	(4.6%)	<0.001	16	(14.7%)	14	(4.3%)	<0.001	0	(0.0%)	1	(0.3%)	>0.999
	Hypomagnesemia	1	(0.9%)	8	(2.4%)	0.300	1	(0.9%)	8	(2.4%)	0.456	0	(0.0%)	0	(0.0%)	-

mCRC, metastatic colorectal cancer; HPN, supplemental HPN program group; Non-HPN, non-supplemental HPN program group; ALT, alanine aminotransferase; AST, aspartate transaminase.

## Data Availability

The data that support the findings of this study are available from the corresponding author, J.Y.W., upon reasonable request.

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
