# Peer review of "Efficacy and Safety of a Parenteral Nutrition Program for Patients with RAS Wild-Type Metastatic Colorectal Cancer Administered First-Line Cetuximab Plus Chemotherapy: A Propensity Score Matching Study"

_nutrients, 2023, doi:10.3390/nu15132971_

Round 1
Reviewer 1 Report
Manuscript is interesting and well-written, however, some issues should be addressed:
- in Introduction section, more emphasis should be put on exact gap in the literature this study is filling and scientific contribution to the topic
- figure 1 could be uploaded in bigger size and quality
- more explanation, emphasis and additional supportive information should be put into future clinical perspectives and utilization that can be derived from these results
Author Response
Reviewer 1
We sincerely appreciate your excellent comments. They have been very helpful to us in revising our manuscript so that it will hopefully be suitable for publication in Nutrients.
Query 1. in Introduction section, more emphasis should be put on exact gap in the literature this study is filling and scientific contribution to the topic
Reply 1. In the Introduction section, we have placed more emphasis on highlighting the specific gap in the literature that this study aims to address. We have also provided a clearer explanation of the scientific contribution of this research to the topic. The routine use of PN support in patients with incurable cancer is generally not recommended. The evidence supporting the benefits of PN intervention in cancer patients is limited. However, nutrition support continues to be prescribed for patients with cancer. Malignant disease represents a significant proportion of cases requiring HPN, ac-counting for approximately half of all cases in a large series. Despite the limited evidence, individualized assessments should be conducted for each patient to determine the most suitable approach to nutrition support based on their specific circumstances and needs.
Query 2. figure 1 could be uploaded in bigger size and quality
Reply 2. Regarding Figure 1, we have uploaded a larger size and higher quality version of the figure to improve visibility and clarity.
Query 3. more explanation, emphasis and additional supportive information should be put into future clinical perspectives and utilization that can be derived from these results
Reply 3. In the Discussion section, we have expanded on the future clinical perspectives and utilization that can be derived from the results obtained in this study. We have included additional explanation, emphasis, and supportive information to provide a comprehensive understanding of the potential implications and applications of the research findings.
While nutritional support can offer benefits to specific patients with mCRC, its use should be approached with caution and based on individual patient evaluations. In-discriminate administration of PN to well-nourished or mildly malnourished patients is not advised due to insufficient evidence demonstrating its overall effectiveness in these patient groups. However, for severely malnourished patients or those at high risk of mal-nutrition due to factors such as impaired intake, gastrointestinal obstruction, or severe treatment side effects, PN may be necessary to maintain or restore their nutritional status. In such cases, the potential benefits of nutritional support outweigh the associated risks.
It is important to consider the appropriateness of nutritional intervention for patients with mCRC based on their disease prognosis and potential for a curative outcome or a long disease-free period after cancer treatment. Patients who have a potentially curable disease or can anticipate a significant period free from cancer after treatment may be best suited for consideration of nutritional intervention. Conversely, in situations where mCRC patients are terminally ill and have a limited life expectancy ranging from weeks to a few months, the routine use of PN support is also not recommended. During this stage, the focus of care shifts towards palliative care, prioritizing comfort and symptom management rather than pursuing aggressive nutritional interventions.
Reviewer 2 Report
In this paper, the authors conducted a comprehensive retrospective study using propensity score matching to examine the effectiveness and safety of treatment with and without an additional home parenteral nutrition (HPN) program in patients diagnosed with RAS wildtype metastatic colorectal cancer (mCRC). The study revealed that the implementation of the HPN program indeed resulted in improved effectiveness and safety of the initial treatment, which included cetuximab in combination with chemotherapy, for patients with RAS wildtype mCRC. Furthermore, the HPN program was found to enhance the nutritional status of patients, enabling them to undergo treatment for a longer duration and reducing treatment interruptions. Additionally, the HPN program demonstrated the potential to prevent adverse events and enhance the overall effectiveness of the treatment, leading to improved patient survival rates. As a result, the authors recommend considering the implementation of the supplemental HPN program for specific mCRC patients who are undergoing targeted therapy along with chemotherapy, with the goal of potentially enhancing the effectiveness of cancer care.
It is a detailed and exhaustive study with careful analysis. The manuscript can be accepted in the present form.
Author Response
Reviewer 2
Query 1. It is a detailed and exhaustive study with careful analysis. The manuscript can be accepted in the present form.
Reply 1. Thank you for your positive feedback on our manuscript. We are glad that you found the study detailed, exhaustive, and carefully analyzed. We greatly appreciate your confidence in the present form of the manuscript and your suggestion that it is suitable for publication in Nutrients.
Reviewer 3 Report
The authors investigated the efficacy and safety of a supplemental home parenteral nutrition (HPN) program for patients with wild-type RAS metastatic colorectal cancer receiving cetuximab plus chemotherapy. This retrospective study employing propensity score matching revealed that the HPN program group had a higher metastasectomy rate, longer duration of treatment, and response than the non-HPN program group. Moreover, the HPN program group significantly improved median overall survival while having a higher risk of hematologic and nonhematologic adverse events. The following points should be clarified.
1. Why did HPN have a higher risk of adverse events? Are adverse events occurring as a result of HPN? The authors should discuss this in more detail.
2. The patient with HPN showed a significant improvement in median overall survival. However, I think that HPN is applied to patients with more stable symptoms. What do you think about this point?
Author Response
Reviewer 3
We sincerely appreciate your excellent comments. They have been very helpful to us in revising our manuscript so that it will hopefully be suitable for publication in Nutrients.
Query 1. Why did HPN have a higher risk of adverse events? Are adverse events occurring as a result of HPN? The authors should discuss this in more detail.
Reply 1. Thank you for raising the question regarding the higher risk of adverse events associated with HPN and suggesting a more detailed discussion on this topic. We have taken your suggestion into account and expanded the discussion section to provide a comprehensive explanation of why HPN carries a higher risk of adverse events. We have also addressed whether these adverse events occur as a result of HPN therapy. The updated discussion section provides insights into the potential factors contributing to the increased risk of adverse events and discusses the relationship between HPN and the occurrence of such events.
HPN can improve the quality of life, nutritional status, and functional outcomes in patients with advanced cancer; however, it does not treat the underlying cause of malnutrition. Furthermore, although HPN enables patients to live a more normal life outside the hospital, it is a high-risk and complex therapy associated with both immediate and long-term complications, particularly in patients with advanced cancer. In terms of safety, HPN can be associated with several risks, such as infection, electrolyte imbalance, glucose intolerance, liver dysfunction, and catheter-related complications. Given the limited life expectancy and the patient's overall condition, the risks associated with nutritional support may outweigh any potential benefits.
Query 2. The patient with HPN showed a significant improvement in median overall survival. However, I think that HPN is applied to patients with more stable symptoms. What do you think about this point?
Reply 2. We have incorporated this point into both the patients and methods and discussion sections.
In the “Patients and Methods” section: It's important to note that patients with an estimated life expectancy of less than three to six months were not intervened with the supplemental HPN program. This decision was made considering their limited life expectancy and the potential burden of initiating long-term nutritional support.
In the “Discussion” section: While nutritional support can offer benefits to specific patients with mCRC, its use should be approached with caution and based on individual patient evaluations. Indiscriminate administration of PN to well-nourished or mildly malnourished patients is not advised due to insufficient evidence demonstrating its overall effectiveness in these patient groups. However, for severely malnourished patients or those at high risk of malnutrition due to factors such as impaired intake, gastrointestinal obstruction, or severe treatment side effects, PN may be necessary to maintain or restore their nutritional status. In such cases, the potential benefits of nutritional support outweigh the associated risks.
It is important to consider the appropriateness of nutritional intervention for patients with mCRC based on their disease prognosis and potential for a curative outcome or a long disease-free period after cancer treatment. Patients who have a potentially curable disease or can anticipate a significant period free from cancer after treatment may be best suited for consideration of nutritional intervention. Conversely, in situations where mCRC patients are terminally ill and have a limited life expectancy ranging from weeks to a few months, the use of PN support is generally not recommended. During this stage, the focus of care shifts towards palliative care, prioritizing comfort and symptom management rather than pursuing aggressive nutritional interventions.
Round 2
Reviewer 1 Report
The authors addressed all of the comments and improved the manuscript. I have no further queries.
Reviewer 3 Report
The authors have satisfactorily addressed the points which I noted.